# Evaluating the Genetic Background Effect on Dissecting the Genetic Basis of Kernel Traits in Reciprocal Maize Introgression Lines

**DOI:** 10.3390/genes14051044

**Published:** 2023-05-06

**Authors:** Ruixiang Liu, Yakun Cui, Lingjie Kong, Fei Zheng, Wenming Zhao, Qingchang Meng, Jianhua Yuan, Meijing Zhang, Yanping Chen

**Affiliations:** Provincial Key Laboratory of Agrobiology, Institute of Food Crops, Jiangsu Academy of Agricultural Sciences, Nanjing 210014, China; liuruixiang@jaas.ac.cn (R.L.); meimei_zmj@163.com (M.Z.)

**Keywords:** maize, genetic background effect, CSL analysis, GWAS, breeding, genomic prediction, introgression lines, QTL, QTN, kernel size

## Abstract

Maize yield is mostly determined by its grain size. Although numerous quantitative trait loci (QTL) have been identified for kernel-related traits, the application of these QTL in breeding programs has been strongly hindered because the populations used for QTL mapping are often different from breeding populations. However, the effect of genetic background on the efficiency of QTL and the accuracy of trait genomic prediction has not been fully studied. Here, we used a set of reciprocal introgression lines (ILs) derived from *417F* × *517F* to evaluate how genetic background affects the detection of QTLassociated with kernel shape traits. A total of 51 QTL for kernel size were identified by chromosome segment lines (CSL) and genome-wide association studies (GWAS) methods. These were subsequently clustered into 13 common QTL based on their physical position, including 7 genetic-background-independent and 6 genetic-background-dependent QTL, respectively. Additionally, different digenic epistatic marker pairs were identified in the *417F* and *517F* ILs. Therefore, our results demonstrated that genetic background strongly affected not only the kernel size QTL mapping via CSL and GWAS but also the genomic prediction accuracy and epistatic detection, thereby enhancing our understanding of how genetic background affects the genetic dissection of grain size-related traits.

## 1. Introduction

Since maize is one of the most important cereal crops for ensuring food and nutrition security [1], its grain yield is a crucial agronomic trait. Three major grain size-related traits, kernel length (KL), kernel width (KW), and kernel thickness (KT), are crucial for determining the maize grain size and yield, making them vital for maize breeding [2]. Moreover, these traits are also related to the maize nutrient content [3], easy mechanical seeding, and early seeding vigor [4]. Therefore, genetic dissection of these grain size-related traits could enhance the current understanding of kernel development and facilitate efficient improvement of maize yield.

Multiple studies have investigated the underlying genetic regulatory mechanisms in various maize kernel mutants, including *defective kernel* (*dek*), *embryo specific* (*emb*), *empty pericarp* (*emp*), *endosperm specific* (*end*), *small kernel* (*smk*), *opaque/floury*, and *shrunken* [2,5]. The cloning and functional analysis of these numerous kernel development-related genes have greatly expanded our understanding of the underlying molecular mechanisms of maize kernel development [5]. Unfortunately, most mutants show negative effects, which limits their application in breeding due to the lack of superior allelic variations when using via marker-assisted selection (MAS) [6]. QTL mapping and association mapping are the two main methods for dissecting the genetic architecture of complex quantitative traits. Over recent decades, numerous QTL and quantitative trait nucleotides (QTNs) for kernel traits using various bi-/multi-parental and association mapping populations [7,8] have been published. Despite the numerous QTL and QTNs being reported, only a few QTL influencing kernel size traits have been fine-mapped and cloned, e.g., *qKL1.07* [9], *qKW4.05* [10], *qKW7* [11], *qKW7b* [12], *qKW9* [13], and *qKL9* [14]. However, even fewer well characterized genes/QTL have been successfully used in breeding for enhancing grain yield. The main reasons are that the QTL mapping results largely depend on the genetic background and no QTL have been simultaneously detected in completely different genetic backgrounds in the multi-parent advanced generation inter-cross (MAGIC) populations research [15]. The QTL are mapped in populations that generally differ from breeding populations and cannot be detectable in the breeding population, thereby limiting their application in molecular breeding. Although association mapping using diversity maize panels can identify favorable alleles, it is difficult to directly use them in breeding due to the poor performance of accessions in terms of many important agronomic traits [15]. Therefore, to identify background-independent QTL, integrating QTL mapping with molecular breeding in the same population will largely minimize the effect of genetic background on QTL detection [15].

Genomic selection uses whole genome-wide molecular markers to predict the breeding values of individuals, and it can capture both major and minor effect markers and is efficient for complex traits [8]. In maize, genomic selection has been widely reported to have multiple practical applications, including inbred line prediction [16], hybrid performance prediction [17], and combining ability prediction [18]. Although these demonstrate the potential of genomic selection in assisting the maize breeding program, how the genetic background affects the prediction accuracy for kernel size needs further investigation.

Genetic analysis using reciprocal ILs derived from the same parents would be more valuable to understand these complex traits [19], especially for studying the genetic background effect on QTL detection [20]. In this study, we developed two reciprocal ILs by crossing two elite inbred lines in our maize breeding program. KL, KW, and KT were recorded in four environments, and we obtained high-density genotypic data via genotyping by target sequencing (GBTS). The aims of this study were to: (1) detect QTL related to maize kernel shape traits using both CSL and GWAS methods, (2) compare the genetic background effect on the genetic dissection of kernel shape traits, and (3) investigate the genetic background effect on genomic prediction accuracy of kernel size-related traits. Our results will not only enrich the current knowledge of the genetic background effect on the genetic dissection of kernel shape traits but also provide valuable information for improving the maize grain yield in the breeding program.

## 2. Materials and Methods

### 2.1. Development of Reciprocal Introgression Lines

Two sets of reciprocal ILs were developed from a cross between *517F* (an elite inbred line with the slender grain) and *417F* (a parent of hybrid *Sukeyu51417* with round kernels). The inbred line *517F* was derived from hybrid *Dika517*, and *417F* was a line from the group B germplasm derived from the modern US hybrid *P78599* (PB) [21]. Both inbred lines were used as paternal inbred lines in our breeding. The F_1_ hybrids were simultaneously backcrossed to *517F* and *417F* to produce their respective BC_1_F_1_ generations. The BC_1_F_1_ individuals were then backcrossed with corresponding parents to produce the BC_2_F_1_, with further backcrossing leading to BC_4_F_1_ populations. The BC_4_F_1_ individuals were then self-crossed for four generations, and following the single seed descent method, the BC_4_F_4_ generation was obtained. Finally, two sets of reciprocal ILs were successfully developed. The reciprocal ILs comprised 149 lines (plus the recurrent parent) in the *517F* background (*517F*-ILs) and 85 lines (plus the recurrent parent) in the *417F* background (*417F*-ILs).

### 2.2. Field Experiment and Trait Measurement

A total of 232 reciprocal ILs along with the parents (*517F* and *417F*) were planted in four environments (Hainan 2018, Hainan 2019, Liuhe 2019, and Liuhe 2020), respectively. A randomized complete block design was applied for field experiments. Each genotype was planted in single row, with each plot being 3 m × 0.5 m (L × W) and containing 13 plants. All field management was performed as per the farmer’s practices. Three grain size-related traits, i.e., KL, KW, and KT, were measured by using five harvested ears in the middle of the plot post natural drying, with the average values of ten kernels in the middle ear being used for further data analysis.

### 2.3. DNA Extraction, SNP Genotyping, and Bin Map Construction

Young leaves from five plants in the middle of the line were bulk-harvested for DNA extraction. Genomic DNA of the two parents and the ILs were extracted, and the genotypes of ILs were determined based on SNPs generated from 20K Array genotyping by target sequencing (GBTS) by the Mol Breeding Biotechnology Company (Shijiazhuang, China) [22], and the ZmB73_RefGen_V4 was used as the reference genome (https://download.maizegdb.org/Zm-B73-REFERENCE-GRAMENE-4.0/Zm-B73-REFERENCE-GRAMENE-4.0.fa.gz accessed on 4 May 2023). Finally, a bin map comprising 3469 bins was constructed in the two ILs based on the SNPs as described previously [23].

### 2.4. Data Analysis

The multi-environment trial analysis was conducted using the MEGA-R software [24]. A mixed linear model was used to calculate the best linear unbiased predictors (BLUPs), variance components, and broad-sense heritability. The model used for data analysis was as follows [25]:*Y_ijk_* = µ + *G_k_* + *E_i_* + *R_j_*_(*i*)_ + *EG_ik_* + *ε_ijk_*(1)
where *Y_ijk_* is the observation of the *k*th genotype in the *i*th environment in the *j*th replicate; μ is the overall mean; *G_k_* is the effect of the *k*th genotype; *E_i_* is the effect of the *i*th environment; *R_j_*_(*i*)_ is the effect of the *j*th replication nested on the *i*th environment; *EG_ik_* is the effect of the interaction between the *i*th environment and the *k*th genotype; and *ε_ijk_* is the effect of experimental error. BLUPs across all environments were used for QTL mapping analysis, GWAS, and genomic prediction analyses. Broad-sense heritability across all environments was calculated as follows [25]:(2)h2=σg2σg2+σge2i+σe2ij
where $\sigma_{g}^{2}$ is the genotypic variance; $\sigma_{ge}^{2}$ is the genotype × environment interaction variance; $\sigma_{e}^{2}$ is the error variance; *i* is the number of environments; and *j* is the number of replications in each environment.

First, the reciprocal ILs were treated as the non-idealized chromosome segment substitution lines (CSL), with the main-effect QTL and digenic epistatic QTL being detected using the CSL function [26] in QTL ICIMapping version 4.2 [27] with default parameter. The LOD thresholds for main-effect QTL and epistatic QTL detection were determined by the 1000 permutation tests with average LOD values of 3.50 and 3.13 in *517F*-ILs and 6.93 and 6.85 in *417F*-ILs [28]. Second, to combine the two reciprocal ILs populations, a joint analysis was performed using the BLINK (Bayesian-information and Linkage-disequilibrium Iteratively Nested Keyway) function of GAPIT3 [29]. The BLINK method uses iterations to select the trait-associated markers, which then are fitted as covariates for testing the other markers, thereby being better for controlling false positives than the kinship approach. Therefore, the BLINK method has better statistical power than FarmCPU [30]. Two models, one with population structure and another without it, were used to compare the effects of the population structure on kernel traits-related QTL detection. The Bonferroni multiple test threshold declared the significant bin markers (*p* = 1.4 × 10^−5^). Genomic prediction analysis was conducted using the genomic best linear unbiased prediction (gBLUP) model by using the “sommer” package in R [31]. The model used to implement gBLUP was as follows [32]:*y* = *Xβ* + *Zμ* + *ε*(3)
where *y* is a vector of phenotypes; *X* is the designated matrix for the fixed effects; *β* is the vector of fixed effects; *Z* is a designated matrix for random effects; *μ* is the vector of additive genetic effects for an individual with variance $K\sigma_{\mu }^{2}$ (in which *K* is the genomic relationship matrix), and *ε* is the vector of residual errors with variance $I\sigma_{e}^{2}$. To test the genomic prediction accuracy effect caused by the population stratification of the reciprocal ILs populations, the population structure was considered as the fixed factor (model 1), random factor (model 2), and without (model 3) in the genomic prediction model.

The genomic predictive ability was measured by computing Pearson’s correlation coefficient between the predicted phenotype and observed phenotypes using the five-fold cross-validation approach (one-fifth of the random sampling as testing dataset while the others as training datasets) with 100 replications. The final accuracy was obtained by taking the average of over 100 replicates.

### 2.5. Identification of Common QTL among Reciprocal ILs and Joint Analysis

According to previous research, all QTL within a 20 cM interval were considered a single QTL [33], with the average recombination rates being 1cM/Mb [34]. Therefore, QTL within 20 Mb in chromosome were defined as the common QTL.

## 3. Results

### 3.1. Bin Map of the Reciprocal ILs Population

Among 49,439 polymorphic SNPs between the two parents, we identified a total of 3469 bins based on recombination sites (Appendix A). These were evenly distributed across 10 chromosomes covering the *517F*-ILs and *417F*-ILs maize B73 genomes 12 and 9 times, respectively [35]. The average length of the chromosome segments was 590 Kb and ranged between 20 kb and 2905 kb. The averaged introgression frequencies of *517F*-ILs were 10.6% and ranged from 0.06–85.6%, whereas those of *417F*-ILs averaged 11.3% and ranged from 0.13–44% (Figure 1).

### 3.2. Phenotypic Performances of Reciprocal ILs and Their Parents

The detailed parameters for the grain size-related traits of the parent inbred lines (*517F* and *417F* lines) and the reciprocal ILs are shown in Table 1. The results indicated that *517F* had a smaller grain size than *417F*. This was supported by the lower values of KL (averaging 9.2 mm for *517F* and 10.4 mm for *417F*), KW (averaging 6.9 mm for *517F* and 8.7 mm for *417F*), and KT (averaging 4.6 mm for *517F* and 5.2 mm for *417F*) across the four testing environments. The ILs’ progenies showed phenotypic trends of their recurrent parents. The mean values across the four environments for KL, KW, and KT were 9.1 mm, 7.3 mm, and 4.3 mm in the *517F*-ILs while being 9.6 mm, 8.4 mm, and 5.1 mm in the *417F*-ILs, respectively. We observed transgressive segregation for all three kernel traits in the reciprocal ILs across four environments, which implied that the favorable alleles for the three kernel traits were from both parent inbred lines. It is also notable that the KL and KW showed larger variations in the *517F*-ILs than in the *417F*-ILs, while the KT was almost equal in the two reciprocal ILs (Table 1).

The KW was significantly positively correlated with the KL (*p* = 1.34 × 10^−19^) and KT (*p* = 1.86 × 10^−19^), with correlation coefficients of 0.54 and 0.51, respectively. However, the KT was not significantly correlated with the KL (*p* = 0.19), which indicated that KT has a different genetic basis than KL. All three kernel-shape traits in the reciprocal ILs were significantly different among the different environments, with KW being higher than KL and KT, thus indicating that KL and KT were affected more by the environment than KW.

An analysis of variance (ANOVA) was performed on each of the three kernel shape traits in the reciprocal ILs and the joint data across four environments (location-by-year combinations). The ANOVA results showed that the genotypes, environments, and the interaction between genotype and environment were all highly significant (Table 2). Genotypes (G) explained an average of 24.5 ± 10.4% of the total phenotypic variation in the reciprocal ILs and the joint data set, ranging from 9% for KT in *517F*-ILs to 45% for KW in the joint data set. Environments (E) explained an average of 6.8 ± 6.4% of the total phenotypic variation in the reciprocal ILs and joint data set, ranging from 0.8% for KW in the joint data set to 16% for KL in *417F*-ILs. The G × E interaction explained an average of 33.9 ± 5.2% of the total phenotypic variation in the three data sets, ranging from 25% for KW in the joint analysis to 41.4% for KT in *517F*-ILs. The broad-sense heritability values (calculated by partitioning the variance into genetic and genotype by environment effects) were >0.7 for KW in both *517F*-ILs, *417F*-ILs and joint analysis. However, the KL trait in the *517F*-ILs and joint analysis was >0.7, while it was ~0.5 in *417F*-ILs. In the three data sets, KT was lower than KW and KL, with 0.52, 0.59, and 0.64 in *417F*-ILs, *517F*-ILs, and the joint analysis, respectively. All results showed that the genetic variation of KW was greater and more stable than that of KL and KT. To reduce the environmental effect, the BLUP value of the kernel trait across all four environments was estimated and then used as the value for marker and trait association analysis and genomic prediction.

### 3.3. QTL Affecting Kernel Shape Traits

Based on the bin marker, we identified a total of 51 QTL for three kernel traits (including 14 for KL, 6 for KT, and 31 for KW) in the reciprocal ILs, joint data set, and methods. The most detected QTL had small to moderate additive effects, whereas 27.5% had effects that could explain >10% of the phenotypic variance per QTL (Table 3; Figure 2).

#### 3.3.1. QTL of Kernel Shape Traits Identified in the Reciprocal ILs

For KL, we detected four and seven QTL in *417F*-ILs and *517F*-ILs by the CSL method, respectively (Table 3, Figure 2), including five on chromosomes 5 and 3, two on chromosome 6, and one on chromosomes 1, 4, and 10. Among the four QTL detected in *417F*-ILs, the donor (*517F*) alleles at the *qKL5.1*, *qKL5.3*, and *qKL10* loci decreased KL by an average of 0.16 (ranging from 0.11 to 0.21), whereas it increased the KL by 0.3 at the *qKL3.2* locus. The phenotypic variance explained (PVE) of the four QTL identified in *417F*-ILs averaged 11.3%, ranging from 9.27% to 14.65%, thus indicating that the four QTL were all major QTL. Among the seven QTL detected in *517F*-ILs, the donor (*417F*) alleles at three (*qKL1*, *qKL5.2*, and *qKL6.1*) loci decreased the KL by 0.21, while it increased KL by 0.23 at four (*qKL3.1*, *qKL4*, *qKL5.4,* and *qKL6.2*) loci. The average PVE of the seven QTL was 6.7%, ranging from 4.15% to 10.97%, with only one (*qKL4*) > 10%, thus indicating that most were minor QTL.

For KT, we identified three and two QTL in *417F*-ILs and *517F*-ILs, respectively (Table 3, Figure 2). We detected two QTL on chromosomes 9 and 2, while one was on chromosome 1. Among the three QTL detected in *417F*-ILs, the donor (*517F*) alleles at the *qKT1* and *qKT2.2* loci decreased the KT by 0.20, whereas at the *qKT2.1* locus they increased the KT by 0.14. For the QTL detected in *517F*-ILs, the donor (*417F*) alleles at all loci increased the KT by 0.10. It is notable that the QTL *qKT2.2* was a major KT-related QTL where the PVE was 15.32%, while the others were minor QTL.

For KW, we detected 17 and 10 QTL in *417F*-ILs and *517F*-ILs, which were unevenly distributed on all chromosomes except chromosome 4. There were eight, seven, and four on chromosomes 3, 5, and 10, respectively. Among the 17 QTL detected in *417F*-ILs, *517F* alleles at 53% loci decreased the KW by an average of 0.30 (ranging from 0.13 to 0.63), whereas at the other loci, it increased the KW by 0.29 (ranging from 0.09 to 0.56). The PVE of the 17 QTL ranged from 0.45% to 21.29%, with an average of 4.94%. Although the QTL on chromosome 3 (*qKW3.1*, *qKW3.2*, and *qKW3.3*) were in a hotspot region, the effect of *qKW3.2* was different from *qKW3.1* and *qKW3.3*. Among the 10 QTL detected in *517F*-ILs, the donor (*417F*) alleles at three loci (*qKW1.2*, *qKW5.2*, and *qKW5.4*) decreased the KW by an average of 0.26, while the other seven loci increased the KW by an average of 0.19 (ranging from 0.12 to 0.30). The PVEs of major QTL (*qKW5.2* and *qKW5.5*) were 13.72% and 10.48%, respectively.

Overall, the QTL of KL, KT, and KW totally explained 45.2%, 28.8%, and 84% of the phenotypic variance in *417F*-ILs population, while explaining 47.2%, 15.5%, and 63.8% of phenotypic variance in *517F*-ILs. None of the single main-effect QTL of the kernel traits were detected in both *417F* and *517F* backgrounds, with common QTL of *cqKW3b*, *cqKW5c*, *cqKW9* for KW and *cqKL5* for KL being detected in both genetic backgrounds, thus indicating QTL for kernel shape traits were strongly affected by the genetic background.

#### 3.3.2. QTL in Joint Analysis of the Two Reciprocal ILs for KL, KT, and KW

To test the effect of genetic background caused by the different backcrossed parents on the kernel traits QTL, we performed the joint analysis using the BLINK method implemented in the GAPIT3 software with and without the population structure model. We identified a total of 10 significant QTL in the two models, including three for KL, one for KT, and six for KW (Table 2, Figure 2). As per our hypotheses, 60% of the significant QTL were model-specific, while only 40% of the significant bins (two for KL, two for KW) were detected in the with and without population structure models, thus indicating the kernel shape trait-related QTL interacted with the genetic background, even with only two parental alleles in two different backcross-caused genetic backgrounds. For example, *qKL3.3* for KL, *qKT1.2* for KT, and *qKW2.2* and *qKW5.6* for KW only can be detected without considering the backcrossed parents’ model, while *qKW2.3* and *qKW5.5* for KW only can be detected in the backcrossed parents considered a covariance model. Especially, the *qKT1.2* for KT explained nearly 50% of the total KT phenotypic variant, thus indicating that KT relied more on genetic background than KL and KW. The phenotypic variation explained by each significant bin-marker ranged from 4.1% to 50%, with a mean of 14.4%. It is notable that the phenotypic variant of the significant bin *Bin5_184.070* on chromosome 5 was > 10% in both models, thus indicating the region near the 184.7 Mb on chromosome 5 has a major KL QTL. Therefore, these results indicated that GWAS mainly detected QTL with a relatively large effect for grain size, while the CSL method could identify more QTL with a small effect besides the QTL with a large effect for kernel shape traits.

Here, 90% of loci detected by the joint analysis overlapped with the QTL obtained using the CSL method, which indicated the robustness of the QTL analyses along with the reliability of the identified QTL and that the complementary models could improve the elucidation of the genetic architecture of the kernel shape traits. All 51 QTL can be divided into 13 common QTL based on their physical position, including three for KL, two for KT, and eight for KW. Among these 13 common QTL, we detected 54% in both reciprocal ILs or both with and without population structure models, designated as background independent QTL. These included two for KL (*cqKL3a* and *cqKL5*), five for KW (*cqKW2*, *cqKW3b*, *cqKW5b*, *cqKW5c*, and *cqKW9*) and no common QTL for KT among the four methods. It is notable that we detected five common QTL only in *417F*-ILs or joint analysis without population structure models, including one for KL (*cqKL3b* on chromosome 3), two for KT (*cqKT1*, in the 29.1–36.2 Mb region of on chromosome 1 and *cqKT2* in the 231.1–236.4 Mb region on chromosome 2), and two for KW (*cqKW3a* and *cqKW10*) designated as the *417F* background dependent QTL, while only one for KW (*cqKW5a*) was the *517F* background dependent QTL (Table 2, Figure 2).

#### 3.3.3. Digenic Epistatic QTL in the Reciprocal ILs for KL, KT, and KW

In the *417F*-ILs, we detected six digenic epistatic bin marker pairs for KW, accounting for 1.27–7.22% of phenotypic variances (Table 4), while no significant bin marker pairs were found for KL and KT, with all pairs occurring between one main-effect QTL and one locus. Among them, five pairs decreased the KW, while the pairs between *Bin9_99.947* and *qKW9.1* (*Bin9_124.092*) on chromosome 9 were found to increase the KW. It was notable that four pairs were between the common QTL *cqKW10* and the adjacent loci in the 18.6–21.7 Mb region on chromosome 9, which indicated that the interaction may be important in regulating KW under the *417F* genetic background. We detected three digenic epistatic pairs in the *517F*-ILs, including one for KL and two for KW, thereby accounting for 15.59–18.41% of phenotypic variance, which were much higher than those in *417F* genetic background. Therefore, no digenic epistatic marker pairs were shared by the two genetic backgrounds, thus showing the complex epistatic interactions for the kernel traits in different genetic backgrounds.

### 3.4. Prediction Accuracies of KL, KT, and KW Estimated with the Reciprocal ILs

We conducted three types of genomic prediction: (a) within the ILs (one was within the *417F*-ILs, while another was within *517F*-ILs), (b) crossing the two reciprocal ILs (one was using the *417F*-ILs as the training set to predict *517F*-ILs, and vice versa), and (c) joining the two reciprocal ILs to predict the kernel shape traits with a five-fold cross validation. The result showed big differences between the three types of genomic prediction accuracies (Figure 3). The highest genomic prediction accuracy was the type within the ILs, with those of cross-reciprocal ILs being almost half of those within ILs. The joint type of genomic prediction accuracies was a little less than within the ILs type, thereby showing that the different genetic background caused by backcross with different parents had a large effect on the kernel traits’ prediction accuracy. Considering the clear population structure, we also conducted the genomic prediction in three models: one with a fixed population structure, another with a random structure, and the third without any population structure. We observed no significant difference in each type of genomic prediction accuracy between the three models, which indicated that taking the population structure into the genomic prediction model did not improve the genomic prediction accuracies much. However, we observed a little improvement in the prediction accuracy when *517F*-ILs were used as the training data set for KW instead of *417F*-ILs, while >10% improvement was seen for KT, and the opposite trend was seen for KL. The genomic prediction accuracies of KL within *417F*-ILs were 9% lower than those within *517F*-ILs, while those of KW were 8% higher than those within *517F*-ILs. Therefore, this was consistent with the number of main-effect QTL that were identified within ILs.

## 4. Discussion

### 4.1. Genetic Background Effect on QTL of Kernel Shape Traits

The maize kernel trait is a complex quantitative trait that is coordinately regulated by KL, KW, and KT. Elucidating the genetic background effect of QTL related to kernel shape traits will help reveal the underlying regulatory mechanisms of maize kernel development. In an advanced backcross population, QTL are identified with the lines having similar genetic backgrounds from the recurrent parent, with the background effect on QTL detection being uncovered by comparing the mapping results from two reciprocal IL populations [20]. Most previous mapping studies inferred the existence of genetic background and main-effect QTL interaction via this above approach [19]. GWAS have been widely applied to the genetic analysis for complex traits in family-based populations, including nested-association mapping (NAM) [38] and MAGIC populations [39], thus proven to be a powerful tool for uncovering the basis of key agronomic traits in maize. In this study, not only CSL analysis but also GWAS with and without population structure models were used to estimate the effect of genetic background on QTL for kernel traits in the reciprocal ILs. Both the CSL and GWAS analysis results showed that the genetic background had a strong effect on the genetic dissection of kernel shape traits. For example, only 23.1% of common QTL can be detected in both genetic backgrounds, which indicated that >76% of QTL might be the genetic-background-specific QTL. In the CSL results, the genetic basis of KL in the *417F*-ILs were three major QTL (PVE > 10%) with one medium size QTL, while it was one major QTL with six medium size QTL (Table 2) in the *517F*-ILs, with the distribution of the major QTL on the genome also being totally different. For the KT and KW traits, it was also the same situation. The population structure-containing model detected a lower number of associations compared to the model without it. There was no significant bin marker for KT, but in the model without the population structure, we identified one significant bin marker explaining 50% of the phenotypic variance. All these suggested that population structure affects the significant bin markers, as reported previously [40]. As the population heterogeneity of the joint ILs was caused by the consecutive backcrosses with two different parents, the effects of the population structure might be considered the genetic-background-dependent effects. These results further confirmed that genetic background largely affects the dissection of traits with complex genetic structures [41]. Additionally, it seems that the *417F* genetic background tends to detect more genetic-background-dependent QTL for kernel shape traits than the *517F*. The most important factor might be the different genetic interactions in the *417F* and *517F* genetic backgrounds, which could be further proven by different numbers of digenic epistatic bin marker pairs for KW in the *417F* and *517F* ILs (Table 4). In addition, the different grain sizes of the two parents resulted from long-term selection in different maize breeding programs or areas, which might generate a different or more complex genetic background. Therefore, a comprehensive evaluation of our results revealed the clear influence of the different genetic backgrounds on QTL detection for kernel shape traits.

In this study, the QTL mapping method of CSL detected more QTL than the GWAS, thus implying that the CSL method was better at detecting QTL with the target trait, which might be due to the low genetic background noise of ILs [42] and the involved recurrent parent analysis [26] that improved the power of QTL mapping. The multiple backcrosses with different parents involved in developing the reciprocal ILs resulted in the totally different frequencies of alleles of the two parents in the reciprocal ILs and a clear population structure when the two reciprocal ILs were joined together. Furthermore, with the stringent Bonferroni correction threshold, it resulted in reduced detection of QTL with small effects in the genome-wide association analysis method.

For the digenic epistasis, we identified six and two pairs of digenic interactions for KW in the *417F*-ILs and *517F*-ILs, respectively, which reflected the significant effect of the genetic background on epistasis detection for kernel traits. The digenic pairs were detected between the main-effect QTL and the bin locus, with similar results being found previously [41]. Although QTL for the grain size-related traits might exist at these bin markers, they were not significant in the QTL analysis due to their small effects. However, such bin locus might contribute to the maize kernel traits.

### 4.2. Comparing the QTL Detected in this Study with Previously Reported QTL

Among the 14 KL QTL we detected in the reciprocal ILs using CSL and GWAS, six QTL covered or were adjacent to the previously reported KL QTL region in maize (Table 3). For example, *qKL1*, *qKL5.2*, *qKL5.3*, *qKL5.4*, *qKL5.5*, and *qKL10.2* overlapped with the previously reported GWAS significant SNP or meta-QTL region, *KL-gCL1-3*, *KL-gCL5-3*, *KL-qCL5-1*, *KL-gCL5-4*, and *KL-gCL10-1* [2], respectively. For KT, three of the six QTL overlapped with the previously studied QTL regions, e.g., *qKT1.1* overlapped with *KT-qCL1-3* and *KT-gCL1-1*, *qKT9.2* overlapped with *KT-qCL9-2*, and especially, *qKT1.2* overlapped with *KT-qCL1-5* and near the significant SNP *S1_35756298* [36], which were detected only in BLINK without the population structure model explaining 50% phenotypic variance. Among the 31 QTL for KW, 16 overlapped with previously reported QTL region. For example, *qKW1.2*, *qKW3.2*, *qKW3.5*, *qKW3.8*, *qKW5.2*, *qKW5.7*, *qKW5.8*, *qKW7*, *qKW10.1*, *qKW10.2*, *qKW10.3*, and *qKW10.4* were co-located with the previously reported KW meta-QTL region [2]. Furthermore, we found *qKW2.1*, *qKW2.4*, *qKW5.1*, and *qKW9.2* adjacent to the previously reported KW meta-QTL region [37]. Therefore, all the above QTL affecting the grain size identified in this study will need further verification via fine mapping and cloning.

### 4.3. Genetic Background Effect on Genomic Selection Accuracy

Genomic prediction is effective for improving complex traits in maize [43]. In each genomic selection type in the present study, the prediction accuracies of kernel shape traits estimated within the reciprocal ILs were moderate, whereas those estimated within introgression lines were higher than those estimated jointly, with one IL being used as the training set to predict the phenotype of another IL. Therefore, we observed lower prediction accuracies between the reciprocal ILs, which indicated that the genetic background is also important in determining the genomic prediction accuracies of kernel-related traits. Previous studies also showed that no single genomic selection model had better performance in all cases due to different backgrounds in training and testing populations, different traits, and different experimental designs [44]. Therefore, the training set for predicting kernel size performance needs to be phenotyped across the different backcrossed breeding populations to eliminate the effect of genetic background on achieving good prediction accuracy when backcrossed genes from GenBank accessions were crossed into elite lines with the genomic selection method.

### 4.4. Implications in Maize Breeding

The efficiency of marker-assisted backcross introgression of QTL from a donor line into a recipient inbred line depends on the stability of QTL [19]. Identification of genetic-background-independent QTL is essential, as the kernel traits are sensitive to genetic background. As compared with genetic-background-dependent QTL, the genetic-background-independent QTL can be applied more easily for genetic improvement as their functions do not depend on the background [42]. Understanding the genetic background and main-effect QTL interactions can also help breeders in deciding which QTL to use in their breeding programs while tailoring the maize hybrid for optimized mechanical seeding. The significantly improved lines and the stable QTL identified in this study are valuable resources for gene discovery and yield improvement. For instance, we detected genetic-background-independent QTL such as *cqKL3a* and *cqKL5* for KL and *cqKW2*, *cqKW3b*, *cqKW5b*, *cqKW5c*, and *cqKW9* in the different genetic backgrounds, which therefore could be used to improve the grain yield indirectly by improving the KL and KW. The introgression of the *517F* alleles at *cqKL3a* and *cqKL5* into the *417F* background by MAS could increase the ratios of dry weight of grain to cob, thus probably resulting in improved grain yield.

Apart from the genetic-background-independent QTL, the genetic-background-dependent QTL can also enhance or counteract phenotypes when MAS is applied in a special genetic background. If the genetic-background-dependent QTL involve numerous loci, the introgression alleles from one background to a different genetic background elite inbred line may generate disappointing results, as the effect would probably disappear with repeated backcross generations. However, if the genetic-background-dependent QTL involve a single locus, the effect of these QTL is conditioned by the allele at the other locus, thus ensuring that a simultaneous introgression might be necessary for getting the desired effect [45]. Moreover, the near isogenic lines developed in this study that accumulated *cqKW10* for KW in the *417F* genetic background, along with the interaction locus, could be used to validate the interaction.

Additionally, for the epistatic pairs identified in this study, any two ILs harboring different QTL/genes controlling the same trait can provide valuable information on how to select the favorable alleles and allele combinations in maize molecular breeding. Furthermore, the prominent allele combinations existing in the special background could be kept in the process of QTL pyramiding breeding, thereby ensuring that the target genetic improvement can be realized quickly [42].

## 5. Conclusions

In this study, we developed two reciprocal ILs from a cross between two elite inbred lines used in breeding programs and then evaluated the grain size in multiple environments. Thereafter, we identified 51 QTL for kernel shape traits using the CSL and GWAS methods and found that they were clustered into 13 common QTL based on their physical position, including seven genetic-background-independent QTL, six genetic-background-dependent QTL, and nine digenic epistatic marker pairs. Therefore, the results demonstrated that genetic background strongly affected the QTL mapping for both the CSL and GWAS methods, along with the genomic prediction accuracy. Thus, the results obtained will not only improve the current understanding of the genetic background effect on dissecting the genetic basis underlying maize grain size but also provide valuable information for future maize breeding programs.

## Figures and Tables

**Figure 1 genes-14-01044-f001:**
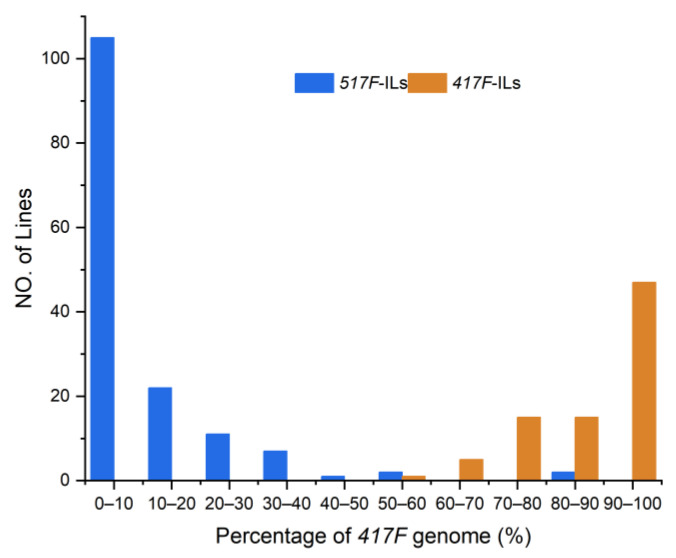
Frequency distribution of *417F* genome in the reciprocal ILs derived from *517F* × *417F*.

**Figure 2 genes-14-01044-f002:**
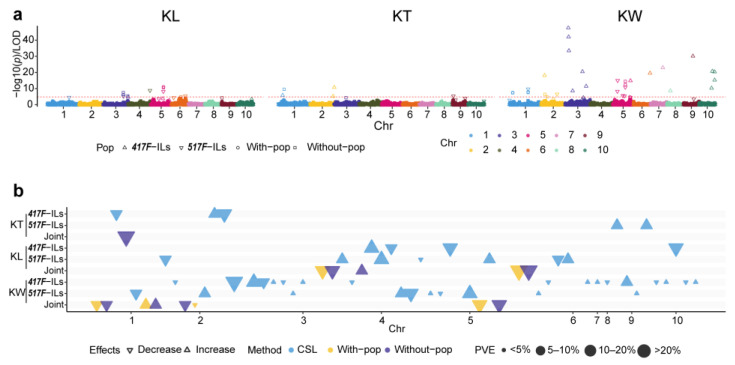
Overview of main effect QTL detected in the reciprocal ILs for grain-size related traits: (**a**), manhattan plot of QTL for kernel traits using CSL and BLINK with/without population structure model; the red dash line indicates the Bonferroni multiple test threshold; (**b**), colored triangles indicate QTL identified by CSL and BLINK with/without the population structure model; the upward triangles indicate an increase in the trait, whereas the downward triangles indicate the opposite effect. The size of the circles indicates the phenotype variation explained by the QTL.

**Figure 3 genes-14-01044-f003:**
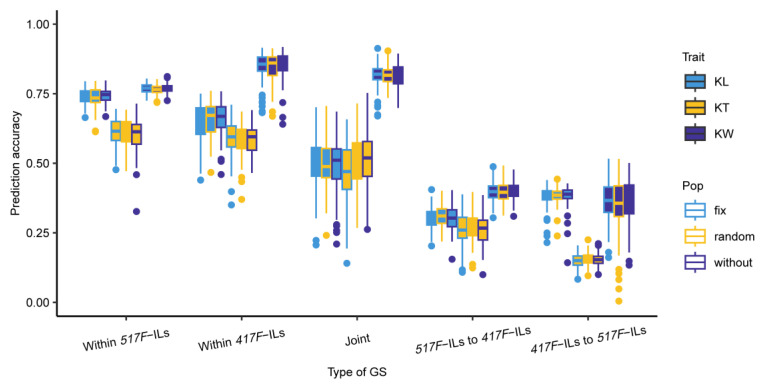
Genomic prediction accuracy of kernel-size related traits in the reciprocal ILs.

**Table 1 genes-14-01044-t001:** Performances of kernel size traits of the reciprocal ILs and their parents *517F* (P1) and *417F* (P2) in four environments.

Traits (mm)	Env	Parents		*517F*-ILs			*417F*-ILs		
		P1	P2	Mean ± SD	Range	CV(%)	Mean ± SD	Range	CV(%)
KL	E1	10.17	11.10	9.81 ± 0.87	7.71–12.07	8.85	10.27 ± 0.71	8.21–11.77	6.96
	E2	8.59	10.03	9.07 ± 1.18	5.44–11.88	13.01	9.8 ± 0.80	7.91–11.46	8.13
	E3	10.36	10.06	9.16 ± 1.09	6.04–11.32	11.86	9.40 ± 0.97	6.78–12.06	10.27
	E4	8.08	10.25	8.46 ± 1.03	5.75–10.79	12.22	9.04 ± 0.93	6.50–11.26	10.31
KW	E1	6.76	9.83	7.30 ± 0.88	5.61–9.85	12.00	8.83 ± 0.74	6.76–10.43	8.39
	E2	6.66	8.05	7.22 ± 0.80	5.61–9.47	11.08	8.35 ± 0.70	6.10–10.05	8.37
	E3	7.01	8.25	7.34 ± 0.75	5.79–9.32	10.29	8.32 ± 0.68	6.65–10.40	8.14
	E4	7.28	8.81	7.26 ± 0.66	5.21–8.81	9.07	8.24 ± 0.78	6.34–10.36	9.51
KT	E1	4.30	5.78	4.33 ± 0.52	3.24–5.99	12.10	4.99 ± 0.59	3.71–6.84	11.82
	E2	3.96	4.96	4.79 ± 0.59	2.85–6.40	12.40	5.20 ± 0.61	3.37–6.75	11.78
	E3	4.63	4.94	4.74 ± 0.68	3.64–6.85	14.32	5.17 ± 0.79	3.79–6.90	15.21
	E4	4.95	5.00	4.75 ± 0.66	3.56–6.40	13.92	5.17 ± 0.79	3.84–7.33	15.22

Note: KL, kernel length (mm); KW, kernel width (mm); KT, kernel thickness (mm); E1, Hainan, 2018; E2, Liuhe, 2019; E3, Hainan,2019; E4, Liuhe, 2020.

**Table 2 genes-14-01044-t002:** Variance components and broad-sense heritability of kernel size-related traits in the reciprocal ILs and joint data set.

Pop	Traits	Variance Components				*h* ^2^
		$\sigma_{g}^{2}$	$\sigma_{env}^{2}$	$\sigma_{ge}^{2}$	$\sigma_{err}^{2}$	
*417F*-ILs	KL	0.23 ***	0.24 ***	0.60 ***	0.43	0.54
	KW	0.24 ***	0.05 ***	0.27 ***	0.34	0.70
	KT	0.14 ***	0.01 **	0.36 ***	0.36	0.52
*517F*-ILs	KL	0.55 ***	0.27 ***	0.67 ***	0.36	0.73
	KW	0.27 ***	0.002 *	0.29 ***	0.37	0.72
	KT	0.06 ***	0.04 ***	0.20 ***	0.34	0.59
Joint	KL	0.49 ***	0.25 ***	0.66 ***	0.38	0.71
	KW	0.54 ***	0.01 *	0.30 ***	0.35	0.83
	KT	0.18 ***	0.02 ***	0.27 ***	0.35	0.64

Note: $\sigma_{g}^{2}$, genotypic variance; $\sigma_{env}^{2}$, environment variance; $\sigma_{ge}^{2}$, genotype by environment interaction variance; $\sigma_{err}^{2}$, error variance; *h*^2^, broad-sense heritability; *, **, and *** significant at *p* < 0.05, 0.01, and 0.001 respectively.

**Table 3 genes-14-01044-t003:** Main-effect QTL affecting kernel shape traits by separate and joint analysis in the reciprocal ILs derived from the cross between *517F* and *417F* with BLUP value across four environments.

Traits	Common QTL ^a^	M-QTL ^b^	Chr ^c^	Bin Marker	CSL ^d^		Joint ^e^		Overlap with Previous Study
					*417F*-ILs	*517F*-ILs	Model with Pop	Model without Pop	
					LOD/Add/PVE (%)	LOD/Add/PVE (%)	−log10(*p*)/Add/PVE (%)	−log10(*p*)/Add/PVE (%)	
KL		*qKL1*	1	*Bin1_225.031*		4.57/−0.22/5.28			*KL-gCL1-3* [2]
	*cqKL3a*	*qKL3.1*	3	*Bin3_206.728*			6.30/−0.15/11.95	7.59/−0.18/10.46	
	*cqKL3a*	*qKL3.2*	3	*Bin3_207.567*		5.24/0.18/5.90			
	*cqKL3b*	*qKL3.3*	3	*Bin3_229.734*				5.43/0.15/6.92	
	*cqKL3b*	*qKL3.4*	3	*Bin3_230.707*	4.64/0.30/14.65				
		*qKL4*	4	*Bin4_236.121*		8.80/0.26/10.97			
		*qKL5.1*	5	*Bin5_8.028*	3.13/−0.15/9.27				
		*qKL5.2*	5	*Bin5_144.103*		3.70/−0.21/4.15			*KL-gCL5-3* [2]
	*cqKL5*	*qKL5.3*	5	*Bin5_179.183*	3.70/−0.21/11.11				*KL-qCL5-1* [2]
	*cqKL5*	*qKL5.4*	5	*Bin5_183.049*		8.19/0.26/9.84			*KL-gCL5-4* [2], *KL-qCL5-1* [2]
	*cqKL5*	*qKL5.5*	5	*Bin5_184.070*			11.06/−0.25/29.35	10.49/−0.24/20.65	*KL-gCL5-4* [2], *KL-qCL5-1* [2]
		*qKL6.1*	6	*Bin6_62.894*		4.34/−0.22/5.01			
		*qKL6.2*	6	*Bin6_152.563*		5.42/0.21/5.99			
		*qKL10.2*	10	*Bin10_142.974*	3.37/−0.11/10.16				*KL-gCL10-1* [2]
KT	*cqKT1*	*qKT1.1*	1	*Bin1_29.052*	5.85/−0.17/7.13				*KT-gCL1-1* [2], *KT-qCL1-3* [2]
	*cqKT1*	*qKT1.2*	1	*Bin1_36.171*				9.54/−0.10/50.00	*KT-qCL1-5* [2] *S1_35756298* [36]
	*cqKT2*	*qKT2.1*	2	*Bin2_231.109*	5.22/0.14/6.32				
	*cqKT2*	*qKT2.2*	2	*Bin2_236.423*	10.74/−0.23/15.32				
		*qKT9.1*	9	*Bin9_7.716*		5.30/0.11/8.88			
		*qKT9.2*	9	*Bin9_147.413*		3.97/0.09/6.64			*KT-qCL9-2* [2]
KW		*qKW1.1*	1	*Bin1_22.486*			7.57/−0.19/9.47	7.32/−0.19/8.80	
		*qKW1.2*	1	*Bin1_206.512*		9.70/−0.23/8.14	7.94/0.17/7.59	7.51/0.17/5.05	*KW-qCL1-4* [2]
	*cqKW2*	*qKW2.1*	2	*Bin2_14.332*	18.10/−0.25/2.78				*MQTL_GW_8* [37]
	*cqKW2*	*qKW2.2*	2	*Bin2_15.379*				6.46/−0.17/5.90	
	*cqKW2*	*qKW2.3*	2	*Bin2_31.026*			4.86/−0.14/4.12		
		*qKW2.4*	2	*Bin2_190.258*		6.51/0.19/5.60			*MQTL_GW_11* [37]
	*cqKW3a*	*qKW3.1*	3	*Bin3_12.511*	47.62/−0.63/21.29				
	*cqKW3a*	*qKW3.2*	3	*Bin3_13.777*	41.97/0.56/15.60				*KW-qCL3-5* [2]
	*cqKW3a*	*qKW3.3*	3	*Bin3_15.625*	33.45/−0.41/9.09				
		*qKW3.4*	3	*Bin3_56.351*	8.56/0.14/1.03				
	*cqKW3b*	*qKW3.5*	3	*Bin3_181.586*	20.50/−0.23/3.50				KW-qCL3-8 [2]
	*cqKW3b*	*qKW3.6*	3	*Bin3_188.137*		3.63/0.12/2.84			
	*cqKW3b*	*qKW3.7*	3	*Bin3_189.058*	4.30/0.09/0.45				
		*qKW3.8*	3	*Bin3_210.382*	11.44/−0.15/1.48				*KW-gCL3-2* [2]
	*cqKW5a*	*qKW5.1*	5	*Bin5_30.612*		8.27/0.30/6.84			*MQTL_GW_25* [37]
	*cqKW5a*	*qKW5.2*	5	*Bin5_36.708*		14.98/−0.36/13.72			*KW-gCL5-2* [2]
	*cqKW5b*	*qKW5.3*	5	*Bin5_167.222*		5.64/0.17/4.7			
	*cqKW5b*	*qKW5.4*	5	*Bin5_175.956*		5.40/−0.19/4.96			
	*cqKW5b*	*qKW5.5*	5	*Bin5_182.560*		10.81/0.25/10.48	12.43/−0.27/15.77		
	*cqKW5b*	*qKW5.6*	5	*Bin5_183.049*				14.43/−0.29/16.05	
	*cqKW5c*	*qKW5.7*	5	*Bin5_208.649*		4.50/0.15/3.78			*KW-qCL5-4* [2]
	*cqKW5c*	*qKW5.8*	5	*Bin5_208.770*	14.86/−0.31/2.07				*KW-qCL5-4* [2]
		*qKW6*	6	*Bin6_165.776*	19.54/0.26/3.33				
		*qKW7*	7	*Bin7_144.792*	22.95/0.37/4.16				*KW-qCL7-5* [2]
		*qKW8*	8	*Bin8_20.414*	8.58/−0.13/1.06				
	*cqKW9*	*qKW9.1*	9	*Bin9_124.092*	30.09/0.38/7.10				
	*cqKW9*	*qKW9.2*	9	*Bin9_133.648*		3.50/0.14/2.78			*MQTL_GW_40* [37]
	*cqKW10*	*qKW10.1*	10	*Bin10_137.330*	10.26/−0.14/1.32				*KW-gCL10-3* [2]
	*cqKW10*	*qKW10.2*	10	*Bin10_139.438*	20.70/0.21/3.69				*KW-gCL10-3* [2]
	*cqKW10*	*qKW10.3*	10	*Bin10_145.998*	20.38/−0.26/3.76				*KW-gCL10-3* [2]
	*cqKW10*	*qKW10.4*	10	*Bin10_146.294*	15.26/0.21/2.30				*KW-gCL10-3* [2], *KW-qCL10-1* [2]

Note: ^a^, QTL within 20 Mb in chromosome; ^b^, main effect QTL; ^c^, chromosome; ^d^, chromosome segment lines analysis; ^e^, joint analysis in model with and without population structure.

**Table 4 genes-14-01044-t004:** Digenic epistatic QTL pairs affecting gain size traits in the reciprocal ILs.

Traits	Bin Marker 1	Bin Marker 2	LOD Aa ^a^	LOD Total ^b^	PVE aa (%) ^c^	PVE Total (%) ^d^	Add1 ^e^	Add2 ^f^	Add by Add ^g^	Pop
KL	*Bin3_196.231*	*Bin3_207.567*	7.46	7.75	10.11	15.98	0.02	0.22	0.1545	*517F*-ILs
KW	*Bin1_206.512*	*Bin1_215.108*	7.01	11.19	3.89	15.59	−0.34	0.09	−0.1232	*517F*-ILs
KW	*Bin5_181.969*	*Bin5_182.560*	7.78	11.53	3.23	18.41	−0.09	0.33	−0.1392	*517F*-ILs
KW	*Bin3_56.351*	*Bin7_34.740*	8.52	10.53	0.29	1.27	0.10	−0.02	−0.0885	*417F*-ILs
KW	*Bin9_18.667*	*Bin10_137.330*	7.56	12.17	0.22	1.57	−0.02	−0.20	−0.0827	*417F*-ILs
KW	*Bin9_18.667*	*Bin10_139.438*	7.50	22.61	0.19	4.08	−0.02	0.15	−0.0819	*417F*-ILs
KW	*Bin9_21.715*	*Bin10_137.330*	7.38	11.88	0.22	1.55	−0.02	−0.20	−0.0822	*417F*-ILs
KW	*Bin9_21.715*	*Bin10_139.438*	7.31	22.32	0.18	4.06	−0.02	0.15	−0.0813	*417F*-ILs
KW	*Bin9_99.947*	*Bin9_124.092*	7.92	30.70	0.44	7.22	−0.09	0.47	0.0847	*417F*-ILs

Note: ^a,b^ LOD score for additive by additive and total; ^c,d^ phenotype variation explained by additive by additive and total; ^e,f^ estimated additive effect of the first and second locus; ^g^, estimated additive by additive effect.

## Data Availability

Data are contained within the article or Appendix A.

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
