# Peer review of "Evaluating the Genetic Background Effect on Dissecting the Genetic Basis of Kernel Traits in Reciprocal Maize Introgression Lines"

_genes, 2023, doi:10.3390/genes14051044_

Round 1
Reviewer 1 Report
Reviewer’s report for manuscript ID: Genes-2353993
Dear Editors of Genes Journal,
The manuscript entitled “Evaluating the genetic background effect on dissecting the genetic basis of kernel traits in reciprocal maize introgression lines” reports the effect of genetic background on the efficiency of QTLs and accuracy of trait genomic prediction on grain size-related traits in maize. This would be very useful information for enhancing our understanding of how the genetic background affects the genetic dissection of grain size-related traits. The method used to get to the conclusion was appropriate and the MS is well written so it was easy to read. However, the MS could be improved with some editions/corrections which I recommend below to be suitable for publication:
1. General comments:
a. A few things which will apply to all sections of the MS such as:
· The use of QTLs is not correct because QTL = Quantitative trait loci which is in plural form already so we don’t need “s”. In case of singular form, it is also QTL but it’s read “locus”.
· There are way too many abbreviations in the MS which make it really difficult to understand and irritating to readers. Pls try to avoid that, some less frequently used or two-word abbreviations should be spelled out at all times, such as GP or GB or GS. With the copy and paste functions in Microsoft-work, this job should be very easy to do.
b. In the Result, details numbers were presented in a big chunk of text (for example, 3.3.1. QTLs of kernel shape traits identified in the reciprocal ILs). This can be translated into a table or a graph (a bar chart or of a box plot) to make them clearer and easier to follow.
c. Comparisons were done in almost all parts of the MS but P values were missing (for example, 3.1. Bin map of the reciprocal ILs population). It would be useful to have the P values in brackets wherever you mentioned something (significantly) bigger/higher/greater/lower….
Table 2 needs P values and explanations on “***” and “*”
2. Specific comments:
a. Line 53: Pls elaborate more on “largely depend on the GB” to demonstrate the importance of the current paper/study.
b. Line 56: What does this mean “accessions in terms of many important agronomic traits”?
c. Line 135: “the BLINK method has better statistical power than other methods” what is/are other methods??
d. Line 150: Pls explain a bit on “the five-fold 150 cross-validation approach”
e. Line 243, Figure 2 caption: “Size of the triangle indicate the phenotype variation explained by the QTL.”. They are circles in Figure 2 instead, pls fix
f. Line 275 & 371: change “special” to “specific”
g. Line 414: change “QTL region” to “QTL regions”
h. Line 474: I can’t see anywhere in the MS information on “based on the physical position” pls add the information or remove this.
The MS was well written so its easy to follow.
Thank you for that
Author Response
Response to Reviewer 1 Comments
Point 1: 1.General comments a. A few things which will apply to all sections of the MS such as: The use of QTLs is not correct because QTL = Quantitative trait loci which is in plural form already so we don’t need “s”. In case of singular form, it is also QTL but it’s read “locus”.
Response 1: All of the “QTLs” in the MS were corrected as “QTL”, except in the reference.
Point 2: There are way too many abbreviations in the MS which make it really difficult to understand and irritating to readers. Pls try to avoid that, some less frequently used or two-word abbreviations should be spelled out at all times, such as GP or GB or GS. With the copy and paste functions in Microsoft-work, this job should be very easy to do.
Response 2: All of the two-word abbreviations (GP, GB, and GS) were spelled out at all times in the MS.
Point 3: b. In the Result, details numbers were presented in a big chunk of text (for example, 3.3.1. QTLs of kernel shape traits identified in the reciprocal ILs). This can be translated into a table or a graph (a bar chart or of a box plot) to make them clearer and easier to follow.
Response 3: The part of “3.3.1. QTLs of kernel shape traits identified in the reciprocal ILs” describes the results of CSL method. It is did a little bit difficult to follow this part and it may be easier to follow when translated into a table or a graph. But all of the information had presented in the Table 3 and Figure 2, if subset the information to form new table or graph , it is more like repeatedly content. Second, as this is part of the results for drawing the conclusions, it would be more better for the whole MS with one table and figure while the readers only need focus on less tables and figures.
Point 4: c. Comparisons were done in almost all parts of the MS but P values were missing (for example, 3.1. Bin map of the reciprocal ILs population). It would be useful to have the P values in brackets wherever you mentioned something (significantly) bigger/higher/greater/lower….
Response 4: The P values of comparisons mentioned significant were added in in brackets.
Point 5: Table 2 needs P values and explanations on “***” and “*”
Response 5: The P values and explanations of “*”, “**”, and “***” of Table 2 were added.
Point 6: 2.Specific comments: a. Line 53: Pls elaborate more on “largely depend on the GB” to demonstrate the importance of the current paper/study.
Response 6: “And no any QTL simultaneously detected in completely different genetic background in the MAGIC populations research” was added after“largely depend on the GB” to further elaborate the large genetic background effect on QTL mapping.
Point 7: b. Line 56: What does this mean “accessions in terms of many important agronomic traits”?
Response 7: Most accessions in the maize germplasm resource (for example tropical maize) always had poor performance in many important agronomic traits, such as grain yield and flowering time, due to the adaptation barriers of the exotic germplasm.
Point 8: c. Line 135: “the BLINK method has better statistical power than other methods” what is/are other methods??
Response 8: As described in the cited paper, the method is FarmCPU which was developed by the same research group previously. To be more precise, “the BLINK method has better statistical power than other methods” in the MS was corrected as “the BLINK method has better statistical power than FarmCPU”.
Point 9: d. Line 150: Pls explain a bit on “the five-fold 150 cross-validation approach”
Response 9: A bit explanation of “the five-fold cross-validation approach”, “one-fifth of the randomly sampling as testing dataset while the others as traing dataset” was added in the bracket.
Point 10:e. Line 243, Figure 2 caption: “Size of the triangle indicate the phenotype variation explained by the QTL.”. They are circles in Figure 2 instead, pls fix
Response 10: The Figure 2 caption: “Size of the triangle indicate the phenotype variation explained by the QTL.” was corrected as “Size of the circles indicate the phenotype variation explained by the QTL.”
Point 11: f. Line 275 & 371: change “special” to “specific”
Response 11: The two “special” words were corrected to “specific”.
Point 12: g. Line 414: change “QTL region” to “QTL regions”
Response 12: The “QTL region” was corrected to “QTL regions”.
Point 13: h. Line 474: I can’t see anywhere in the MS information on “based on the physical position” pls add the information or remove this.
Response 13: The clean sequencing data needs to align to the reference B73 genome (https://download.maizegdb.org/Zm-B73-REFERENCE-GRAMENE-4.0/Zm-B73-REFERENCE-GRAMENE-4.0.fa.gz) when genotyping the reciprocal ILs by the 20K Array genotyping by target sequencing (GBTS). Each of SNPs and bin markers were named by the chromosome name with physical position. The reference genome information was added in the part of “2.3. DNA extraction, SNP genotyping, and bin map construction”.
Reviewer 2 Report
I found this paper to be very interesting and it has shown how the markers can be different in two genetic backgrounds. This is perhaps not unexpected but the size of the difference could be unexpected.
Overall it is very well written with only a few English edits necessary.
Line 178 “the favor alleles”, should be “the favorable alleles”.
Line 343 should be better with … one was with a fixed population structure, another was random, and the third model was without any population structure.
Line 360 QTLS were identified within lines…
The endlish language is very good with only a few edits required, which I have suggested.
Author Response
Response to Reviewer 2 Comments
Point 1: Line 178 “the favor alleles”, should be “the favorable alleles”.
Response 1: “the favor alleles” was corrected as “the favorable alleles”.
Point 2: Line 343 should be better with … one was with a fixed population structure, another was random, and the third model was without any population structure.
Response 2: Line 343 was changed to “one was with a fixed population structure, another was random, and the third model was without any population structure”.
Point 3: Line 360 QTLS were identified within lines…
Response 3: “QTLS were identified within lines” was corrected as “QTL were identified within ILs”.